# In Vitro, Molecular Docking and In Silico ADME/Tox Studies of Emodin and Chrysophanol against Human Colorectal and Cervical Carcinoma

**DOI:** 10.3390/ph15111348

**Published:** 2022-10-31

**Authors:** Wasim Ahmad, Mohammad Azam Ansari, Abdulrhman Alsayari, Dalia Almaghaslah, Shadma Wahab, Mohammad N. Alomary, Qazi Mohammad Sajid Jamal, Firdos Alam Khan, Abuzer Ali, Prawez Alam, Abozer Y. Elderdery

**Affiliations:** 1Department of Pharmacy, Mohammed Al-Mana College for Medical Sciences, Dammam 34222, Saudi Arabia; 2Department of Epidemic Disease Research, Institute for Research and Medical Consultations (IRMC), Imam Abdulrahman Bin Faisal University, Dammam 31441, Saudi Arabia; 3Department of Pharmacognosy, College of Pharmacy, King Khalid University, Abha 61421, Saudi Arabia; 4Complementary and Alternative Medicine Unit, College of Pharmacy, King Khalid University, Abha 61421, Saudi Arabia; 5Department of Clinical Pharmacy, College of Pharmacy, King Khalid University, Abha 61421, Saudi Arabia; 6National Centre for Biotechnology, King Abdulaziz City for Science and Technology (KACST), Riyadh 11442, Saudi Arabia; 7Department of Health Informatics, College of Public Health and Health Informatics, Qassim University, Al Bukayriyah 52741, Saudi Arabia; 8Department of Stem Cell Research, Institute for Research and Medical Consultations (IRMC), Imam Abdulrahman Bin Faisal University, Dammam 31441, Saudi Arabia; 9Department of Pharmacognosy, College of Pharmacy, Taif University, Taif 21944, Saudi Arabia; 10Department of Pharmacognosy, College of Pharmacy, Prince Sattam Bin Abdulaziz University, Al-Kharj 11941, Saudi Arabia; 11Department of Clinical Laboratory Sciences, College of Applied Medical Sciences, Jouf University, Sakaka 72388, Saudi Arabia

**Keywords:** in silico, emodin, chrysophanol, colorectal cancer, cervical carcinoma

## Abstract

Anthraquinones (AQs) are present in foods, dietary supplements, pharmaceuticals, and traditional treatments and have a wide spectrum of pharmacological activities. In the search for anti-cancer drugs, AQ derivatives are an important class. In this study, anthraquinone aglycons chrysophanol (Chr), emodin (EM) and FDA-approved anticancer drug fluorouracil were analyzed by molecular docking studies against receptor molecules caspase-3, apoptosis regulator Bcl-2, TRAF2 and NCK-interacting protein kinase (TNIK) and cyclin-dependent protein kinase 2 (CDK2) as novel candidates for future anticancer therapeutic development. The ADMET SAR database was used to predict the toxicity profile and pharmacokinetics of the Chr and EM. Furthermore, in silico results were validated by the in vitro anticancer activity against HCT-116 and HeLa cell lines to determine the anticancer effect. According to the docking studies simulated by the docking program AutoDock Vina 4.0, Chr and EM had good binding energies against the target proteins. It has been observed that Chr and EM show stronger molecular interaction than that of the FDA-approved anticancer drug fluorouracil. In the in vitro results, Chr and EM demonstrated promising anticancer activity in HCT-116 and HeLa cells. These findings lay the groundwork for the potential use of Chr and EM in the treatment of human colorectal and cervical carcinomas.

## 1. Introduction

Uncontrolled malignant neoplasms, the hallmark of cancer, are a prominent cause of mortality in both developed and developing nations. There are several medications on the market that may slow tumor growth, but none of them are completely effective or safe [1]. Colon cancer (CC) is the world’s third leading cause of cancer-related deaths. CC is very common in older people, a major public health issue in every country. By 2030, the worldwide prevalence of CC is expected to increase by 60%, with over 2.2 million new cases and 1.1 million deaths [2]. Lymph node involvement has a significant impact on cervical cancer survival. Cervical cancer is the fourth most common type of cancer in women, accounting for 6.6% of all female cancers in 2020, with 604,000 new cases expected [3]. Therefore, cancer is a significant concern for medical doctors and researchers involved in this area. However, with the use of adjuvant and neoadjuvant treatments such as surgery and radiation, as well as chemotherapy, survival rates have grown significantly. Even though these drugs can be very effective, they don’t work very well because tumor cells are very sensitive to chemotherapeutic bioactive and have a lot of side effects [4]. As a result, finding efficient adjuvant treatments and alternate therapy medications has been a major focus of cancer treatment research.

AQs have been utilized in traditional medicine for many centuries, and the aloe plant is one of the most well-known examples [5]. AQs are also present in foods, dietary supplementation, and pharmaceuticals, and possess a wide spectrum of pharmacological activities. Anthraquinone analogues are a class of aromatic chemicals that include anthraquinone as their primary structural nucleus [6]. For millennia, anthraquinones have been utilized as laxatives. Extracts of *Rheum palmatum*, *Polygonum cuspidatum*, and *Aloe vera* are used to produce emodin and chrysophanol, a natural anthraquinone molecule with anti-tumor potential [7]. Emodin possesses multiple biological effects, including antibacterial, antiviral, diuretic, anti-inflammatory, vasorelaxant, antiulcerogenic, and anticancer properties. In vitro and in vivo investigations have demonstrated that emodin has a broad spectrum of efficacy against cardiovascular illnesses, including antioxidant, anti-inflammatory, myocardial cell fibrosis inhibiting, and other pharmacological properties [8,9]. According to an in vivo study, emodin is safe for usage in both male and female mice when administered at doses of 20–80 mg/kg for 12 weeks [10]. However, some pharmacological, pharmacokinetic, and toxicity studies in vitro and in vivo have shown that emodin may also have adverse effects on the reproductive system, the kidneys, and the liver, especially when used over an extended period of time [11]. In a study [12], chrysophanol was found to be the most abundant free anthraquinone molecule [8]. In the literature, the pharmacological applications of chrysophanol are also well documented as it may have the potential for the prevention and treatment of a variety of diseases such as asthma, osteoporosis, cancer, Alzheimer’s disease, atherosclerosis, diabetes and diabetic complications, retinal degeneration, osteoarthritis, and atopic dermatitis. Chrysophanol has a protective effect against cerebral ischaemia–reperfusion (I/R) due to its anti-inflammatory properties [13]. Chrysophanol has a variety of biological properties such as neuroprotective, antidiabetic, hepatoprotective, anti-ulcer, anti-cancer, anti-inflammatory, antiviral, antibacterial, and antifungal actions [14]. In clinical practice, members of the anthraquinone family are utilized as anticancer medicines and anthracyclines [15]. Several anticancer drugs contain the anthraquinone moiety as their core component. AQ-based medicines like valrubicin, doxorubicin, idarubicin, epirubicin, and mitoxantrone have been successfully utilized to treat hematological and solid cancers. As a result, the AQs core remains a promising framework for the development of innovative therapeutic options [6]. A range of AQ analogs, including chrysophanol, emodin, and several synthetic analogs, have been shown to have increased anticancer activity against a variety of targets, including telomerase, topoisomerase, kinase, matrix metalloproteinases, DNA, and various cell cycle stages [16].

Recent research has shown that multitarget treatments outperform single-target therapeutics in terms of efficacy, toxicity, and drug resistance [17]. The process of developing a new treatment is expensive and time-consuming, and scientists are seeking to solve the riddle by repurposing or using molecules from natural sources, which gives innovative possibilities to find new treatments [18]. Most of the prior research that has looked at colon and cervical cancers on these anthraquinone analogues through in vivo and in vitro investigations has not come to any definitive conclusions which can explore crucial cancer-related protein targets through in silico investigations. Furthermore, there is a paucity of information on the binding processes and interaction energies of these anthraquinone analogues with CDK2, TNIK, apoptosis regulator Bcl-2, and caspase-3 targets. On the other hand, to the best of our knowledge, Chr and EM have received a very little amount of attention in terms of molecular docking in silico studies against cancer-related protein targets responsible for the progression of cervical and colon carcinomas. Therefore, in the current research, molecular docking and in silico, in vivo investigations were carried out to evaluate the drug candidacy of Chr and EM as compared to the FDA-approved anticancer drug fluorouracil to develop more effective antifolate medications against cervical and colon cancer. Furthermore, the docking results were confirmed by in vitro experiments which demonstrated the tumor inhibitory effects of Chr and EM on HCT-116 and HeLa cell lines.

## 2. Results

### 2.1. Molecular Docking and I -Silico Analysis

Using the AutoDock, EM and Chr compounds and the FDA-approved anticancer drug fluorouracil were docked onto the active sites of four protein molecules: caspase-3, Apoptosis regulator Bcl-2, TNIK, and CDK2, in order to obtain insight into the anticancer mechanism of action of these components. 3D models of EM and Chr were retrieved from the PubChem Database (Figure 1). This docking investigation was conducted using the 3D crystal structures of cyclin-dependent kinase 2 (PDB:6GUE), TNIK (PDB:2X7F), apoptosis regulator Bcl-2 (PDB:4MAN) and caspase-3 (PDB:4QU8) generated by Discovery Studio Visualizer 2021. The crystal structures of all selected target proteins have been shown in Figure 2. The protein molecule’s putative binding sites and the search space’s configuration were discovered. Visual assessment of the results revealed that EM and Chr were entrenched within the target protein’s ATP binding region. The active site of the chosen target proteins was occupied by screened compounds EM and Chr. Figure 3A,B shows the docking results of EM and receptors caspase-3, which represents 2D and 3D graphical representation of amino acid residues involved in hydrophobic interaction and different color dotted lines showing different types of bonding including hydrogen bonds formation during emodin and caspase-3 protein interaction. Binding affinity of emodin with reference ligands caspase-3 showed three H-bond interactions (shown by green broken lines) as CYS 163, HIS 121, and ARG 207. Interacting residues involved in hydrophobic interaction were SER205, GLN161, ARG64, ALA162, SER120, GLY122, and TYR204.

Figure 4A,B showed the 2D and 3D graphical representation of amino acid residues involved in hydrophobic interaction and different types of bonding, including hydrogen bond formation during EM and apoptosis regulator Bcl-2 interaction. We detected no H-bond interactions during the binding affinity of EM with reference ligands, Bcl-2. Aromatic rings of test compounds exhibited nonpolar interactions (indicated by the broken pink and purple lines), such as LEY 134 and MET 112. While the residues involved in hydrophobic interaction were ALA146, PHE150, GLU149, VAL153, PHE109, PHE101, TYR105, ASP108, VAL130, and GLU133.

Further, Figure 5A,B and Figure 6A,B show 2D and 3D interactions of EM with human TNIK and CDK2, respectively. Three H-bond interactions were observed in the binding affinity of Emodin to the reference ligand TNIK, whereas the binding affinity of emodin to CDK2 showed two H-bonds such as ASP 86 and LEU 83. The docking results of EM with receptor molecules caspase-3, apoptosis regulator Bcl-2, TNIK, and CDK2 have been summarized in Table 1, which shows that EM was predicted to bind to the active-site cavity of the TNIK with a binding score of approx. −8 kcal/mol, which was higher than CDK2 ligands (−6.55 kcal/mol) and caspase-3 (−7.42 kcal/mol). EM showed the lowest (−7.42 kcal/mol) binding to the active-site cavity of the Bcl-2.

Additionally, Figure 7A,B and Figure 8A,B represent the 2D and 3D interaction of Chr with caspase-3 protein and Apoptosis regulator Bcl-2, respectively. Two H-bond interactions have been shown by the Chr with reference ligands caspase-3 binding as compared to Chr with reference ligands Bcl-2, which showed no H-bond interactions. The 2D and 3D interaction of Chr with human TNIK and CDK2 have been shown in Figure 9A,B and Figure 10A,B, respectively, which shows two and one H-bond interaction in the binding affinity of Chr to the reference ligand TNIK interacting protein kinase and with CDK2, respectively. Table 2 summarizes the docking data for Chr with the receptor molecules caspase-3, Bcl-2, TNIK, and CDK2 which shows that Chr was predicted to bind to the active-site cavity of the TNIK with a binding score of −8.25 kcal/mol, which was higher than EM. CDK2 ligands and Chr had a binding score of (−7.71 kcal/mol) and caspase-3 (−7.37 kcal/mol), while the same compound had the lowest binding score (−6.83 kcal/mol) to the active-site cavity of the Bcl-2.

Furthermore, molecular docking analysis was performed to compare the scoring functions obtained after docking with one of the FDA-approved anti-cancer drugs fluorouracil to the selected receptor molecules, namely caspase-3, apoptosis regulator Bcl-2, TNIK, and CDK2 Figure 11A,B, and Figure 12A,B depict the anti-cancer drug fluorouracil’s 2D and 3D interactions with the caspase-3 protein and the apoptosis regulator Bcl-2. Seven H-bond interactions have been shown by fluorouracil with caspase-3 binding protein as compared to Bcl-2, which showed five H-bond interactions. 2D and 3D interactions of fluorouracil with human TNIK and with CDK2 have been shown in Figure 13A,B and Figure 14A,B respectively, which show three and five H-bond interactions. Table 3 summarizes the docking data for fluorouracil with the receptor molecules caspase-3, Bcl-2, TNIK, and CDK2, which shows that fluorouracil was predicted to bind to the active-site cavity of the TNIK with a binding score of −3.89 kcal/mol, which is lesser than EM (–8.15 kcal/mol) and Chr (−8.25 kcal/mol). Docking results of fluorouracil with receptor molecules caspase-3, Apoptosis regulator Bcl-2, TNIK and CDK2 have been summarized in Table 3. Based on obtained free energy, the comparative analysis of molecular docking analysis of emodin, chrysophanol and the FDA-approved anti-cancerous drug fluorouracil is shown in Table 4, and it has been observed that EM and Chr shows stronger molecular interaction than that of the FDA-approved anticancer drug fluorouracil (Table 4).

### 2.2. Pharmacokinetic Parameters of Emodin and Chrysophanol

The present work used the SwissADME webserver to investigate the probable pharmacokinetic features of the EM and Chr compounds. The result of ADME prediction from the SwissADME webserver is shown in Table 5 and Table 6. Furthermore, the online tool pkCSM server was used to predict the additional toxicity features of selected compounds. Toxicity prediction data obtained from the pkCSM server is summarized in Table 7. These metrics provide the evaluation of drug absorptivity of particular compounds through the cell membrane as well as the possibility for hydrogen bonding to their respective targets. The GI absorption of compounds was high, with a skin permeation value of −5.34 to −6.02, respectively, for Chr and EM. According to the drug-likeness prediction rule, the molecular weight of a compound should be between 150 and 500 g/mol. Chr and EM showed a molecular weight of 254.24 and 270.24. Total polar surface area (TPSA) should be between 20 and 130 Å^2^, and it was found to be 74.60 and 94.83, respectively, for Chr and EM. The average of all predicted Log po/w should not be higher than six, which was found for emodin 1.87 and for Chr 2.38. The bioavailability score for each compound was 0.55, which should not be less than 0.25. The maximum tolerated dose of a compound in humans for possible drug candidates should be ≤ 0.477 log(mg/kg/day). Our results revealed that Chr and EM showed −0.256 and 0.158 log (mg/kg/day) maximum tolerated doses, respectively, which is within the limit. Drug-likeness predictions from the SwissADME server for EM and Chr are shown in Table 5 and Table 6. All of the rules with excellent permeability are qualified by the EM and Chr. As a result, it may be utilized as an effective medicine. In terms of bioactivity, EM and Chr have a larger bioactivity score, which suggests that they have good absorption, metabolism, and distribution. A stronger bioactivity score further indicates that EM and Chr may be powerful inhibitors of nuclear receptors, protease inhibitors, and GPCRs.

The projected ADME results will aid in the development of leads with improved drug-like characteristics. Figure 15 depicts a BOILED-EGG graph of the chemicals EM and Chr. The white component of a BOILED-EGG’s yolk indicates that molecules are passively absorbed by the gastrointestinal system, while the yellow section indicates that chemicals are passively penetrated across the blood–brain barrier (BBB). Red dots indicate compounds that the P-glycoprotein is not expected to remove from the central nervous system. It is apparent from Figure 15 that no chemicals pass through the blood–brain barrier. The ADME characteristics of emodin were the best among both the compounds in this group.

### 2.3. Impact of Emodin and Chrysophanol on Cancer Cells Viability

The effects of EM and Chr on both colon cancer (HCT-116) and cervical cancer (HeLa) cells were examined, and a significant decrease in cell viability after the treatments of EM and Chr was observed. The treatments EM and Chr showed significant inhibitory action on cancer cell growth and proliferation (Figure 16 and Figure 17). The average cell viability was found to be 15.37 and 13.00% at dose of 5 µg/mL of EM against HCT-116 and HeLa cells, respectively, while cell viability after treatment with Chr was 16.37 and 13.00%, respectively, at similar doses. Furthermore, the effects of EM and Chr on non-cancerous cells, i.e., HEK-293, were investigated, and no significant inhibitory effects were found (Figure 16 and Figure 17). Compared to noncancerous HEK-293 cell lines, these results indicate that EM and Chr have a potent inhibitory effect on HCT-116 and HeLa cells.

### 2.4. Effects of Emodin and Chrysophanol on the disintegration of cancer DNA

The number of cancer cells was dramatically reduced after treatment with EM and Chr, as the number of DAPI stained cells was significantly lower in the Chr- and EM-treated cells (Figure 18B,C) compared to control cells (Figure 18A). The reduction in the number of cancer cells could be attributed to apoptosis, or programmed cell death.

## 3. Discussion

Anthraquinone, also known as 9,10-dioxoanthracene, is a class of natural and synthetic chemicals with a wide range of biological functions. In recent years, there has been a great deal of interest in the synthesis of anthraquinone derivatives, as well as in the modification of their structures and the evaluation of their bioactivity. EM and Chr are the most important anthraquinone analogues [19]. In drug development, anthraquinones have become an important family of chemicals due to their wide range of biological characteristics. Using the computational process known as molecular docking, it is feasible to anticipate how a ligand will bond to a protein with a given three-dimensional structure [20]. As a result, it is extremely desirable to investigate the potential of these anthraquinone libraries to discover hit or lead-like compounds for future research. We found less literature on anthraquinone analogue binding mechanisms and interaction energy with caspase-3, apoptosis regulator Bcl-2, TNIK, and CDK2, important cancer-related protein targets. Thus, in this study, we used protein-ligand molecular docking to examine anthraquinone analogs’ EM and Chr binding mechanisms and interaction energies with four major cancer-related protein targets, which are responsible for a wide range of malignancies. Studying the ADMET and effectiveness profiles of a compound might help researchers better comprehend them. Therefore, the AdmetSAR database was used to predict the toxicity profile and pharmacokinetics of EM and Chr. Furthermore, in silico results were validated by the in vitro anticancer activity against HCT-116 and HeLa cell lines to determine the anticancer effect. Almost all drugs have protein targets, and almost all drug metabolizing enzymes (DMEs) are membrane-bound proteins that are essential in signal transduction [20]. Due to lower efficacy, redundancy, undesirable safety characteristics, drug resistance, on-and off-target toxicities, and anti-target and counter-target actions associated with single target therapeutic agents, modern drug discovery concentrates on developing drugs with multiple targets as compared to a single target [17].

TNIK is a promising therapeutic target in the treatment of colorectal cancer, which is characterized by abnormal Wnt-signaling. TNIK inhibition has arisen as an appealing anti-cancer treatment option because of its role in Wnt-mediated colorectal carcinoma [21]. Additionally, in recent years, a large number of naturally occurring chemicals have been cited as starting points for the development of novel drugs functioning as protein kinase inhibitors and, consequently, possible therapeutic candidates [22]. A docking approach was applied in this study for the compounds EM and Chr and the anti-cancerous drug fluorouracil to explore their role as TNIK inhibitors. Chr and EM showed a binding score −8.25 kcal/mol −8.15 kcal/mol, respectively, against protein target TNIK, while the anti-cancerous drug fluorouracil showed a binding score −3.89 kcal/mol. Cyclin-dependent kinases (CDKs) are one class of serine/threonine proteins. These serine/threonine proteins are renowned for their important functions in the regulation of cell divisions [23]. Cancer and other disorders associated with aberrant cellular production can be effectively treated by blocking several enzymes that regulate the cell cycle’s progression [24]. A docking screening procedure was carried out for Chr and emodin compounds in order to uncover some hits that could potentially be promising CDK2 inhibitors. We have found an estimated free energy of binding for CDK2 receptor in the case of EM −7.61 Kcal/mol and Chr −7.71 Kcal/mol, respectively, while for the anti-cancerous drug fluorouracil, the estimated free energy was −5.82Kcal/mol. We studied the interaction of EM and Chr with the caspase-3 protein target by the virtual docking method. As a key player in programmed cell death, caspase-3 degrades several proteins in cells upon activation. Due to the fact that many chemotherapeutic agents have been shown to trigger apoptosis in cancer cells, it has been proposed that targeting apoptosis regulators may be a potential technique for the identification of anticancer drugs [25]. Caspase-3, -8, and -9 expression levels have been proven in previous research to be effective prognostic indicators in tumors of the digestive system, particularly in colorectal cancer [26]. As shown in Table 4, caspase-3 was found to have a high affinity for EM (−7.42 Kcal/mol) and Chr (−7.37 Kcal/mol) when compared to that of the standard anti-cancerous drug fluorouracil (−4.10 Kcal/mol). The molecular docking approach was utilized to evaluate the various binding mechanisms of EM and Chr compounds inside the active region of Bcl-2. Overexpression of the Bcl-2 protein is thought to occur in over half of all clinical malignancies. As a mechanism for the activation of the Bcl-2 gene, chromosomal translocation has been shown to be associated not only with non-Hodgkin lymphomas but also with small cell lung cancer and breast cancer [27]. As apoptotic regulators, the BCL-2 family of proteins has been linked to colorectal cancer (CRC) development, progression, and resistance to treatment [28]. It has been suggested that BCL-2 may play a vital role in a relatively early stage of cervical tumorigenesis [29]. For the physiological Bcl-2, EM and Chr bind firmly to the binding groove, with a calculated binding free energy of −6.55 Kcal/mol for the EM and −6.83 Kcal/mol for Chr, respectively, whereas the binding affinity for the anti-cancerous drug fluorouracil was −2.64Kcal/mol (Table 4).

The active sites of screened compounds EM and Chr were found to be occupied by all the four specified target proteins with good binding energy. The target proteins bind best to the molecular docking score with the lowest. According to our docking data, EM and Chr were the most attractive for cancer targeting proteins TNIK and CDK2, followed by caspase-3. Bcl-2 target legend showed the least interaction with both compounds. Compared to docking scores with all four legends, EM and Chr had a greater estimated free binding energy than the standard anti-cancerous drug fluorouracil (Table 4). In Figure 3, Figure 4, Figure 5, Figure 6, Figure 7, Figure 8, Figure 9, Figure 10, Figure 11, Figure 12, Figure 13 and Figure 14, the ligand–receptor interactions are depicted by dotted lines. There are hydrogen bonds, which are green in color, and hydrophobic interactions, which are pink in color. Finally, emodin was determined to be the greatest hit for the target protein’s active site pocket based on the docking score and hydrogen bond analyses. The researchers hypothesize that non-covalent interactions with the active region of the protein could result in the target being inhibited or blocked. This finding further establishes EM and Chr anticancer potential, which may assist in advancing this biologically active molecule to the next stage of medication research and development.

Emodin and chrysophanol’s drug-like properties are mostly derived from the SwissADME online tool and the pkCSM webserver. Lipinski’s rule of five-described drug-like qualities such as rotatable bonds, molecular weight, H-bond donors-acceptors, and log P value. As per Lipinski’s rule of five, any drug-like compound must have a molecular weight of 500, H-bond acceptors with a value of 10 and 5, H-bond donors with a value of 5, rotatable bonds with a value of 10, and a partition coefficient (log P) value of 5. EM and Chr appear to meet all five of Lipinski’s drug-likeness criteria, which are presented in Table 3 and 4. Both the selected AQ derivatives passed the Veber’s rule, Ghose filter, unweighted quantitative estimate of drug-likeness (QED), weighted quantitative estimate of drug-likeness (QED), and blood–brain barrier (BBB) likeness rule. All of the foregoing findings point to these compounds as being a promising treatment agent for cancer.

Furthermore, in this study, the in vitro anti-cancer activities of EM and Chr were examined against HCT-116 and HeLa cell lines to validate the in-silico results. The cytotoxicity was assessed using the MTT assay, and the inhibitory concentration (IC_50_) values were determined after 48 h of treatment. An analysis of the cells’ viability revealed a decline in the percentage of viable cancer cells. Additionally, the effects of EM and Chr on non-cancerous cells (HEK-293) were studied, and no significant inhibitory effects were found (Figure 16 and Figure 17). These data suggest that EM and Chr have a considerable inhibitory impact on HCT-116 and HeLa cells compared to non-cancerous HEK-293 cell lines. It has been reported that emodin was found to cause apoptosis and reduce the viability of colon cancer cells in a time-and dose-dependent manner [30]. Previously, Chen et al. reported that aloe-emodin inhibits cell growth and promotes apoptosis in several cancer cells [31]. Most of the studies reported the in vitro anticancer activity of EM and Chr, which has not been widely studied in colorectal and cervical carcinoma. Autophagy causes apoptosis in colon cancer cells via generating ROS, which strengthens the importance of the autophagy pathway in the apoptosis process [32]. A previous study showed that emodin induces the death of colon cancer cells by activating caspases, modulating the Bcl-2 protein family, and decreasing mitochondrial membrane potential [30]. A study revealed that EM inhibits the intestinal inflammation associated with carcinogenesis and reduces the initiation and development of AOM/DSS-induced intestinal tumorigenesis [33]. Emodin inhibited the NF-kB pathway and down-regulated BMP2 expression via inhibiting TNF, TNF receptor-associated factors, and p65 protein synthesis [34]. In another study, it has been reported that emodin (20 or 40 mg/kg in vivo and 10–40 µM in vitro) suppressed mTOR/HIF-1/VEGF signaling and reduced inflammation in LPS-induced ALI rats and RAW264.7 cells [35]. According to other research, emodin, in varying concentrations and after varying amounts of time incubated, can inhibit mitotic activity and induce changes in the cytoskeleton, which in turn induces mitotic death in cervical cancer cells. Mitotic death is an alternative form of cancer cell death [36]. It has been suggested that emodin (less than µ125 M) inhibits the growth of PC9, H1650, H1299, H1975, and A549 cell lines. Emodin reduced ILK expression by upregulating MAPK extracellular signaling-regulated kinase (MEK)/ERK1/2 and AMPK signaling pathways, as well as decreasing Sp1 and c-Jun protein expression [37].

In in vivo research, chrysophanol (0.1 mg/kg) was found to inhibit the expression of tumor necrosis factor-a (TNF-a), interleukin-1b (IL-1b), and NF-jB p65 in male C57BL mice [38]. Chi-Cheng et al. [39] first reported that chrysophanol triggered nonapoptotic cell death in J5 human liver cancer cells by boosting ROS production, damaging DNA, mitochondrial malfunction, loss of ATP, and enhancement of lactate dehydrogenase activity. Deng et al. [40] reported that chrysophanol activated the intrinsic mitochondrial apoptotic mechanism to inhibit colorectal cancer cell growth and enhance apoptosis. Also, in vivo studies showed that chrysophanol inhibited tumor development and enhanced apoptotic cells in tumor xenografts without toxicity. Further, proteomic iTRAQ analysis revealed that decorin is the key target of chrysophanol, and it was found that chrysophanol exposure resulted in an upregulation of decorin in the tumor tissues, and ectopic decorin expression significantly enhanced the pro-apoptotic actions of chrysophanol in colorectal cancer cells. Deng et al. [40] concluded that chrysophanol has anti-neoplastic effects on colorectal cancer cells both in vitro and in vivo through regulating decorin, indicating its therapeutic potential for colorectal cancer. A recent study showed that chrysophanol inhibited H/R-induced apoptosis by downregulating the expression of cleaved caspase-3, p-JNK, and Bax, while upregulating the expression of Bcl-2 [14].

The MTT test was used to determine the in vitro antiproliferative activity of anthraquinones and their extracts against four cancer cell lines: MIAPaCa-2, HCT-116, MCF-7, and T47D. The result showed a mechanism of action comprising cell cycle analysis, and assessment of mitochondrial membrane potential (MMP) loss was performed. The extract showed prolonged cell cycle arrest, apoptosis, and the loss of MMP (m) were all seen in concentration-dependent manners when extracts were used [41]. After treatment with EM and Chr, a considerable reduction in the number of cancer cells was seen, as shown by the fact that the number of DAPI-stained cells was much lower in the Chr and EM-treated cells (Figure 18B,C), in comparison to the untreated control cells (Figure 18A). The in vitro anticancer properties of anthraquinone derivatives against four distinct cell lines: PC3, HT-29, HeLa, and HepG2 were reported [19]. All of the aforementioned studies allow us to hypothesize the tumor inhibitory effects of Chr and EM on HCT-116 and HeLa cell lines. According to the findings of our research, EM and Chr have the potential to be used in the treatment of colon and cervical cancer.

In silico molecular docking studies helped us understand how emodin and chrysophanol ligands interact with target proteins. The correlations between in silico and in vitro inhibition findings were strongly connected; the lowest molecular docking score was observed to have the highest binding affinity to the target proteins. Overall, the comparison of docking scores with all the four target proteins revealed that the estimated free binding energy of Chr was comparatively higher than EM, while in vitro cell viability test results have shown that EM and Chr both significantly inhabit proliferation of HeLa and HCT-116 cells, at a concentration of 5 µg/mL, respectively. On the other hand, a considerable reduction in the number of cancer cells was seen, as shown by the fact that the number of DAPI-stained cells was much lower in the Chr and EM treated cells.

## 4. Materials and Methods

### 4.1. Chemical Structure Preparation

Chr and EM structural file information was downloaded from the PubChem database (http://pubchem.ncbi.nlm.nih.gov/compound accessed on 18 January 2022). Corina molecular structure generator tool [42] was used to generate 3-Dimensional (3D) structures, and further CHARMM force fields [43] were applied using Discovery Studio visualizer 2021.

### 4.2. Receptor Molecules Preparation

The information and 3D crystal structure of cyclin-dependent kinase 2 (PDB:6GUE), TRAF2 and NCK-interacting protein kinase (TNIK) (PDB:2X7F), Apoptosis regulator Bcl-2 (PDB:4MAN) and caspase-3 (PDB:4QU8) were accessed and downloaded from Protein Data Bank (PDB). Removal of water molecules and HETATM from native 3D structures and CHARMM force field [43] implementation were done using Discovery Studio visualizer 2021.

### 4.3. Docking Studies

We utilized the AutoDock tool [44] for the screening of prepared compounds Chr and EM on the basis of binding affinity against selected receptor molecules. AutoDock4 provides fast and quick interaction analysis output. The docking methodology was adopted from our previous study [45]. The docking assisted virtual screening was performed on the active site after setting the AutoDock default parameters. The Lamarckian genetic algorithm (LGA) was used for receptor–compound flexible docking calculations [46]. The AutoDock generated 10 conformations of receptors and chemical complexes were analyzed for the interactions of the docked structure using Discovery Studio Visualizer 2021 [44,45].

### 4.4. Drug-Likeness and ADMET

Computational predictions of ADME, drug-likeness, and pharmacokinetics properties of Chr and EM were performed using the SwissADME online tool [47,48,49]. Also, additional toxicity analysis of Chr and EM compounds was performed using the pkCSM online server [50]. The physicochemical parameters and the ADME characteristics, such as the molecular weight (MW), hydrogen bond acceptor (HBA), the topological polar surface area (TPSA), the hydrogen bond donor (HBD), and the octanol/water partition coefficient (LogP) and the rotatable bond count (RB), were determined for EM and Chr compounds. The EM and Chr canonical smiles were submitted to admetSAR software, and results were generated in minutes. The most up-to-date and comprehensive hand-selected data for substances with known ADMET characteristics may be found on the webserver.

#### D Molecular Modeling and Interaction

Screened data and selected natural compound data were further analyzed and 2D and 3D models were generated using Discovery Studio Visualizer 2021.

### 4.5. Evaluation of Anticancer Activity of Emodin and Chrysophanol

#### 4.5.1. MTT Assay

The standard MTT assay was performed to assess the cell viability and antiproliferative activity of EM and Chr [51]. The effects of different concentrations of EM and Chr (2.0 µg to 25 µg/mL) on the cell viability and proliferation of two cancerous cell lines, i.e., human colorectal carcinoma (HCT-116) and human cervical (HeLa), as well as non-cancerous cell lines, i.e., embryonic kidney cells (HEK-293), were investigated as described previously [52]. Briefly, the HCT-116, HeLa, and HEK-293 cells were treated with EM and Chr for 48 h. After that 10 µL of MTT (5mg/mL) solution were added in control (without EM and Chr) and treated groups and were incubated in a CO_2_ incubator for 4 h. After incubation, media were removed and the 96-well culture plate was filled with 1% DMSO. Then, optical density was measured using an ELISA plate reader at 570 nm, and finally, the percentage of cell viability was calculated.

#### 4.5.2. Morphology of Cancer Cell: DAPI Staining Assay

Further, the effects of EM and Chr on the morphology of cancer cells were examined by a DAPI staining assay [52]. Briefly, after 48 h of treatment, treated (40 µg/mL of EM and Chr) and untreated (control) cells were exposed to ice-cold paraformaldehyde (4%) and Triton X-100. Then, cells were stained with DAPI (1.0 μg/mL) for 5 min in dark conditions, and after that, the cells were washed twice with PBS. Finally, the image was captured by using a confocal scanning microscope (Zeiss, Germany).

### 4.6. Statistical Analyses

The statistical studies were carried out with the help of SPSS version 16 for Windows. Comparisons between control and treatment groups were made using *t*-tests. A statistically significant variation was defined as a value of *p* < 0.05.

## 5. Conclusions

Computer-generated screening on huge libraries of chemicals, assessed results, and modules for how the ligands interfere with the target are enormously helpful in finding novel inhibitors through docking. In comparison to the anti-cancer drug fluorouracil, Chr and EM both demonstrated better binding scores against the protein targets caspase-3, apoptosis regulator Bcl-2, TNIK, and CDK2. When the overall docking scores were compared with those of the four legends, the estimated free binding energy of Chr was found to be greater than EM. Server-based ADME analysis yields quick data that could be valuable in the discovery of lead compounds. Findings from a theoretical analysis of the Chr and EM compounds indicated that they are chemically reactive and have drug-like characteristics. In the in vitro results, Chr and EM demonstrated promising anticancer activity in HCT-116 and HeLa cell. These findings set the groundwork for the potential use of Chr and EM in the treatment of human colorectal and cervical carcinoma, and we urge further in vivo research to validate the in vitro and in silico results of EM and Chr against HCT-116 and HeLa cell lines.

## Figures and Tables

**Figure 1 pharmaceuticals-15-01348-f001:**
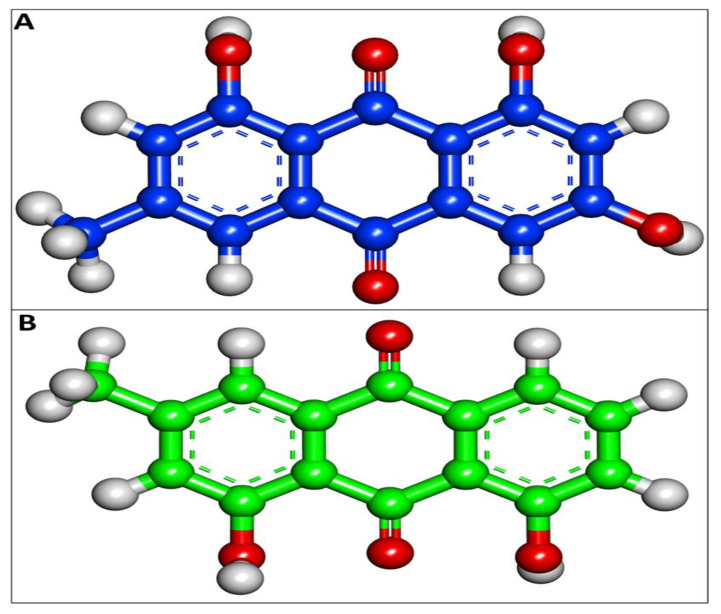
3D structure of (**A**) Emodin PubChem CID:3220 and (**B**) Chrysophanol PubChem CID:10208 generated by using Discovery Studio Visualizer 2021.

**Figure 2 pharmaceuticals-15-01348-f002:**
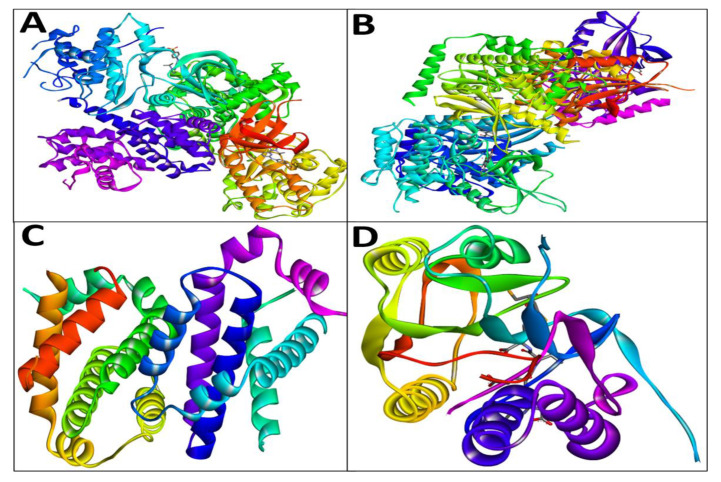
Crystal structure of (**A**) cyclin-dependent kinase 2 (PDB:6GUE), (**B**) TRAF2 and NCK-interacting protein kinase (PDB:2X7F), (**C**) apoptosis regulator Bcl-2 (PDB:4MAN), and (**D**) caspase-3 (PDB:4QU8). 3D graphics were generated using Discovery Studio Visualizer 2021.

**Figure 3 pharmaceuticals-15-01348-f003:**
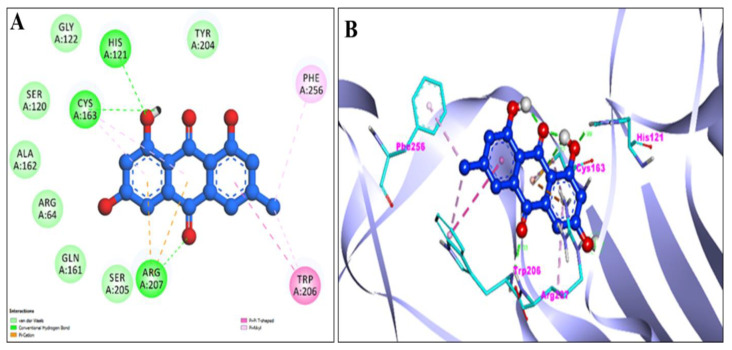
2D (**A**) and 3D (**B**) representation of interaction of emodin (blue color) and caspase-3 protein (purple color with ribbon pattern) (PDB:4QU8).

**Figure 4 pharmaceuticals-15-01348-f004:**
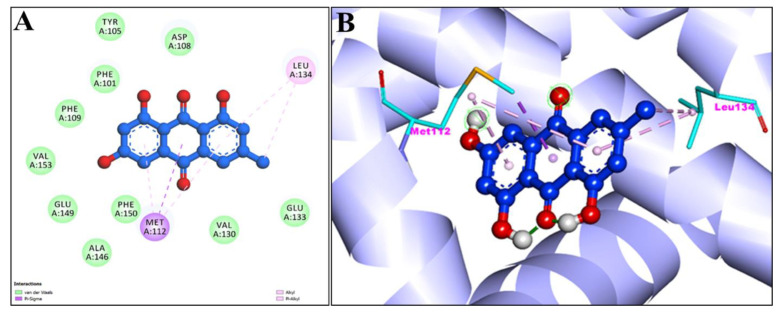
2D (**A**) and 3D (**B**) representation of interaction of Emodin (blue color) and Apoptosis regulator Bcl-2 (purple color with ribbon pattern) (PDB:4MAN).

**Figure 5 pharmaceuticals-15-01348-f005:**
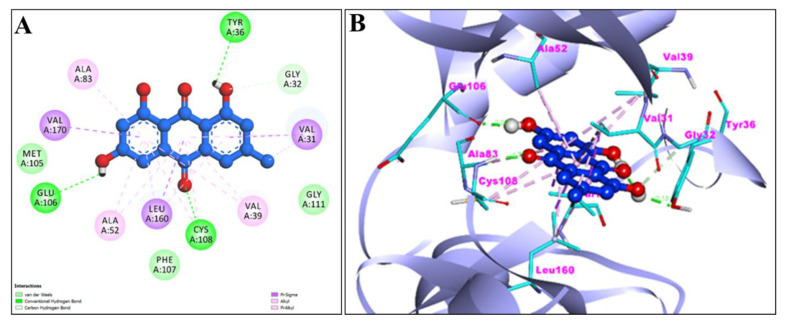
2D (**A**) and 3D (**B**) representation of interaction of emodin (blue color) and human Traf2- and Nck- interacting protein kinase (purple color with ribbon pattern) (PDB:2X7F).

**Figure 6 pharmaceuticals-15-01348-f006:**
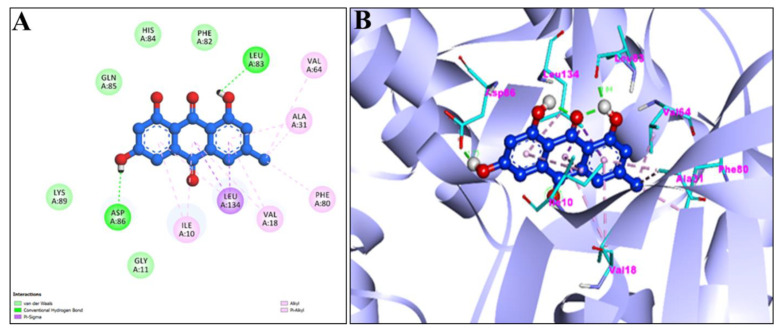
2D (**A**) and 3D (**B**) representation of interaction of emodin (blue color) and cyclin-dependent kinase 2 (purple color with ribbon pattern) (PDB:6GUE).

**Figure 7 pharmaceuticals-15-01348-f007:**
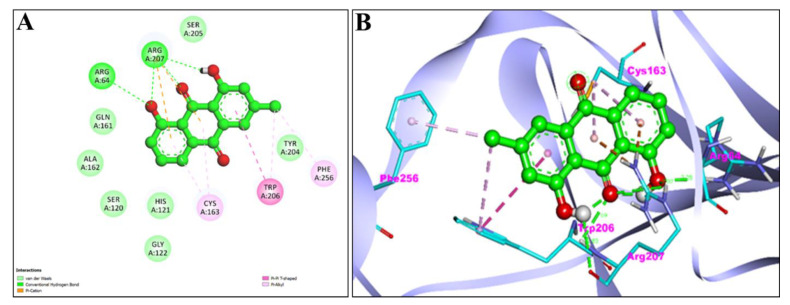
2D (**A**) and 3D (**B**) representation of interaction of chrysophanol (green color) and caspase-3 protein (purple color with ribbon pattern) (PDB:4QU8).

**Figure 8 pharmaceuticals-15-01348-f008:**
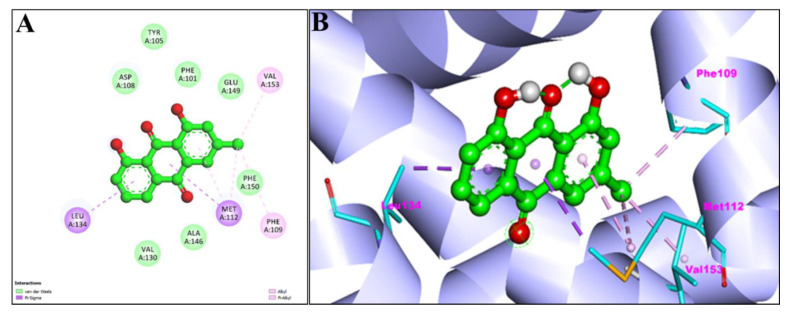
2D (**A**) and 3D (**B**) representation of interaction of chrysophanol (green color) and apoptosis regulator Bcl-2 (purple color with ribbon pattern) (PDB:4MAN).

**Figure 9 pharmaceuticals-15-01348-f009:**
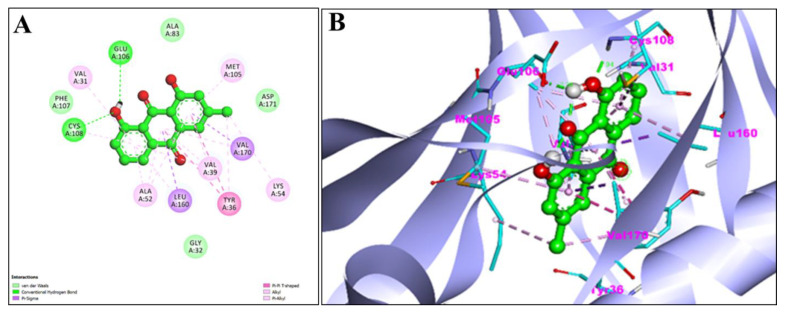
2D (**A**) and 3D (**B**) representation of interaction of chrysophanol (green color) and human Traf2- and NCK-interacting protein Kinase (purple color with ribbon pattern) (PDB:2X7F).

**Figure 10 pharmaceuticals-15-01348-f010:**
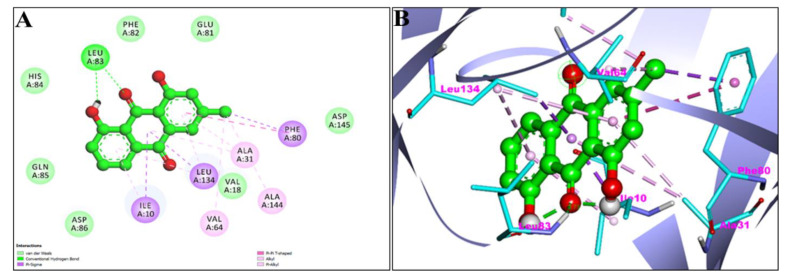
2D (**A**) and 3D (**B**) representation of interaction of chrysophanol (green color) and cyclin-dependent kinase 2 (purple color with ribbon pattern) (PDB:6GUE).

**Figure 11 pharmaceuticals-15-01348-f011:**
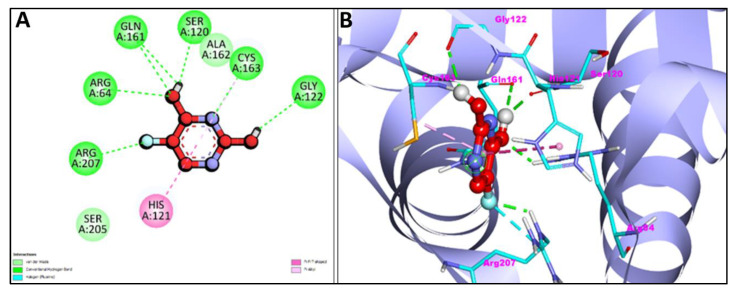
2D (**A**) and 3D (**B**) representation of interaction of anti-cancerous drug fluorouracil and caspase-3 protein (purple color with ribbon pattern) (PDB:4QU8).

**Figure 12 pharmaceuticals-15-01348-f012:**
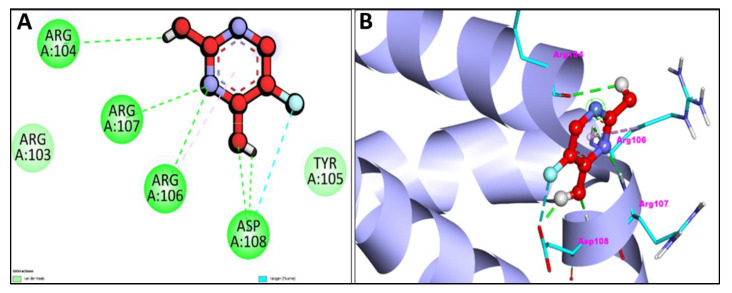
2D (**A**) and 3D (**B**) representation of interaction of anti-cancerous drug fluorouracil and apoptosis regulator Bcl-2 (purple color with ribbon pattern) (PDB:4MAN).

**Figure 13 pharmaceuticals-15-01348-f013:**
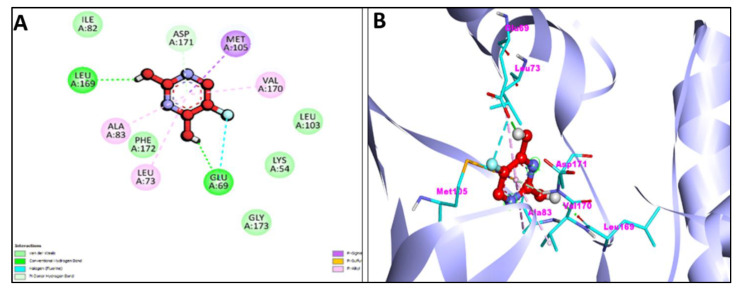
2D (**A**) and 3D (**B**) representation of interaction of anti-cancerous drug fluorouracil and human Traf2- and Nck- interacting protein Kinase (purple color with ribbon pattern) (PDB:2X7F).

**Figure 14 pharmaceuticals-15-01348-f014:**
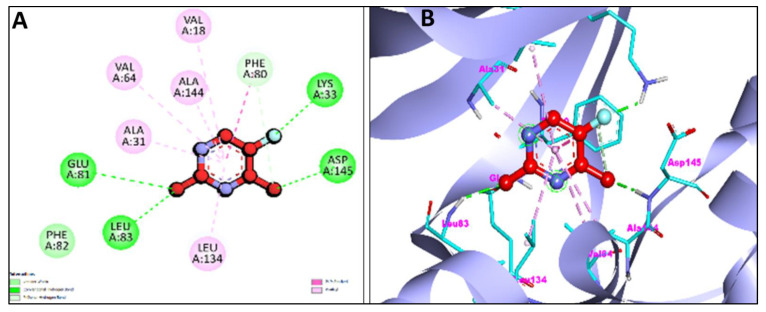
2D (**A**) and 3D (**B**) representation of interaction of anti-cancerous drug fluorouracil and cyclin-dependent kinase 2 (purple color with ribbon pattern) (PDB:6GUE).

**Figure 15 pharmaceuticals-15-01348-f015:**
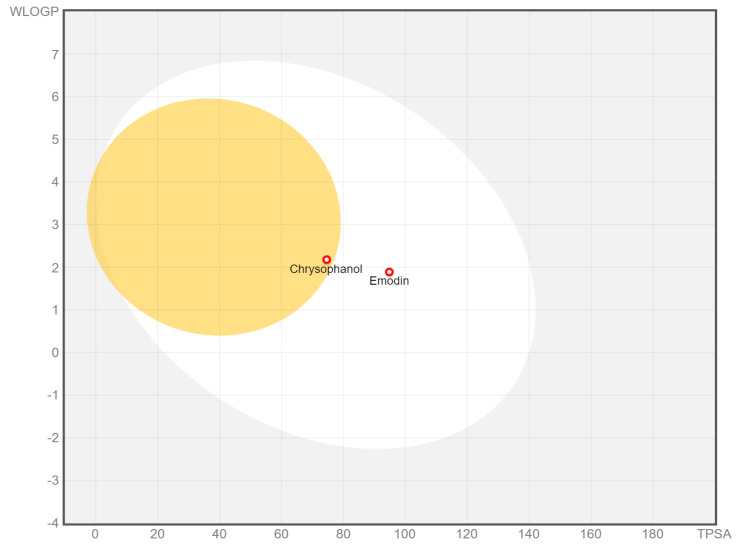
BOILED—EGG graph of compounds Emodin and Chrysophanol. The white component of a BOILED—EGG’s yolk indicates that molecules are passively absorbed by the gastrointestinal tract, while the yellow section indicates that chemicals are passively penetrate across the blood–brain barrier (BBB). Red dots indicate substances that the P-glycoprotein is not expected to remove from the central nervous system.

**Figure 16 pharmaceuticals-15-01348-f016:**
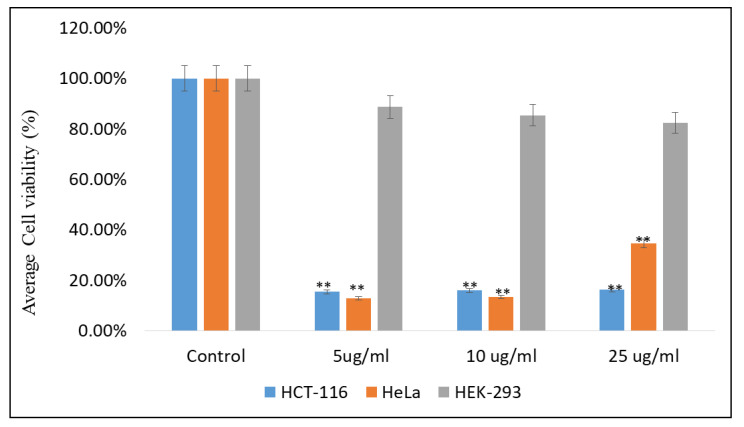
Effects of emodin on HeLa, HCT-116 and HEK-293 cells cell viability assessed by MTT assay. ** *p* < 0.01.

**Figure 17 pharmaceuticals-15-01348-f017:**
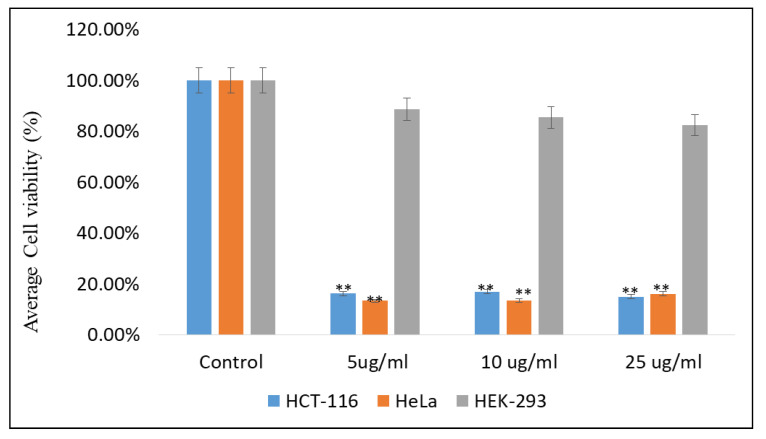
Effects of chrysophanol on HeLa, HCT-116 and HEK-293 cells cell viability assessed by MTT assay. ** *p* < 0.01.

**Figure 18 pharmaceuticals-15-01348-f018:**
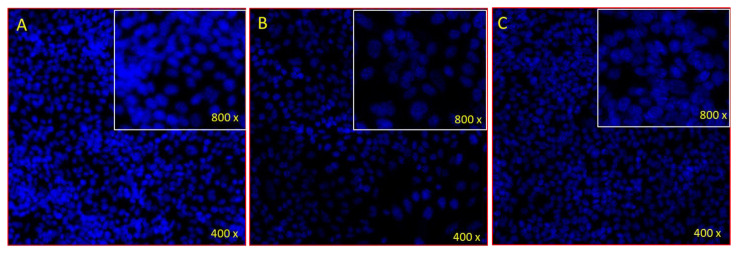
Cancer cells morphology by DAPI staining. (**A**) Control; (**B**) and (**C**) treated with 40 µg/mL of chrysophanol and emodin, respectively.

**Table 1 pharmaceuticals-15-01348-t001:** Showing docking results of emodin with receptor molecules caspase-3, apoptosis regulator Bcl-2, TRAF2 and NCK-interacting protein kinase and cyclin-dependent protein kinase 2.

Receptor Name	Estimated Free Energy of Binding (Kcal/mol)	H-Bond Details	H-Bond Length (A^0^)	Inhibition Constant (Ki)	Interacting Residues Involved in Hydrophobic Interaction (van der Waals)	Alkyl/Pi-Alkyl/Pi-Sigma
Caspase-3 (PDB:4QU8)	−7.42	A:HIS121:HD1-:UNK1:O19	2.03306	3.61 uM	SER205,GLN161,ARG64,ALA162,SER120,GLY122,TYR204	Pi-Alkyl = PHE256 Pi-Pi T-shaped= TRP206 PI-CATION = ARG207
		A:ARG207:HN-:UNK1:O14	2.61505			
		:UNK1:H29-A:CYS163:SG	3.86125			
		A:ARG207:NH1-:UNK1	1.98608			
Apoptosis regulator Bcl-2 (PDB:4MAN)	−6.55	NA	NA	15.69 uM	ALA146,PHE150,GLU149,VAL153,PHE109,PHE101,TYR105,ASP108,VAL130,GLU133	ALKYL/PI-ALKYL = LEU134 Pi-sigma = MET112
TRAF2 and NCK-interacting protein kinase (PDB: 2X7F)	−8.15	A:GLY32:CA-:UNK1:O8	3.26587	1.05 uM	MET105,PHE107,GLY111,	ALKYL/PI-ALKYL = ALA83,ALA52,VAL39 PI-SIGMA = VAL170,LEU160,VAL31
		:UNK1:H30-A:GLU106:O	2.20444			
		:UNK1:H26-A:TYR36:OH	2.13319			
		A:CYS108:HN-:UNK1:O14	1.79344			
Cyclin-dependent protein kinase 2 (PDB:6GUE)	−7.61	:UNK1:H26-A:LEU83:O	1.83759	2.63 uM	GLY11,LYS89,GLN85,PHE82	ALKYL/PI-ALKYL = ILE10,VAL18,PHE80,ALA31,VAL64 PI-SIGMA = LEU134
		:UNK1:H30-A:ASP86:OD2	1.73228			

**Table 2 pharmaceuticals-15-01348-t002:** Showing docking results of chrysophanol with receptor molecules caspase-3, apoptosis regulator Bcl-2, TRAF2 and NCK-interacting protein kinase and cyclin-dependent protein kinase 2.

Receptor Name	Free Energy of Binding (Kcal/mol)	Hydrogen Bond Details	H-Bond Length	Inhibition Constant (Ki)	Interacting Residues Involved in Hydrophobic Interaction (Van der Waals)	Alkyl/Pi-Alkyl/Pi-Sigma
Caspase-3 (PDB:4QU8)	−7.37	A:ARG64:HH21-:UNK1:O19	2.27807	3.97 uM	HIS121,GLY122,SER120,ALA162,GLN161,SER205,TYR204	Pi-Alkyl = PHE256 Pi-Pi T-shaped = TRP206
		A:ARG207:HN-:UNK1:O10	2.58829			
		A:ARG207:HE-:UNK1:O10	2.99886			
		A:ARG207:HE-:UNK1:O19	2.01928			
		:UNK1:H25-A:ARG207:O	2.82811			
Apoptosis regulator Bcl-2 (PDB:4MAN)	−6.83	NA	NA	9.79 uM	ASP108,TYR105,PHE101,GLU149,PHE150,ALA146,VAL130	VAL153,PHE109 Pi-sigma = LEU134,MET112
TRAF2 and NCK-interacting protein kinase (PDB: 2X7F)	−8.25	A:CYS108:HN-:UNK1:O19	1.93541	899.29 nM	PHE107,ALA83,ASP171,GLY32	ALKYL/PI-ALKYL = VAL31,ALA52,VAL39,LYS54,MET105 Pi-Pi T-shaped = TYR36 PI-SIGMA = LEU160,VAL170
		:UNK1:H25-:UNK1:O10	2.08381			
Cyclin-dependent protein kinase-2 (PDB: 6GUE)	−7.71	A:LEU83:HN-:UNK1:O10	2.73142	2.22 uM	GLN85,HIS84,ASP86,PHE82,GLU81,ASP145,VAL18	ALKYL/PI-ALKYL = ALA31,VAL64,ALA144 PI-SIGMA = ILE10,LEU134,PHE80
		:UNK1:H29-A:LEU83:O	1.76833			

**Table 3 pharmaceuticals-15-01348-t003:** Showing docking results of anti-cancerous drug fluorouracil with receptor molecules caspase-3, apoptosis regulator Bcl-2, TRAF2 and NCK-interacting protein kinase and cyclin-dependent protein kinase-2.

Receptor Name	Estimated Free Energy of Binding (Kcal/mol)	Inhibition Constant (Ki)	H-Bond Name	H Bond LENGTH (Angstrom)
Caspase-3 (PDB:4QU8)	−4.10	986.76uM	A:ARG64:HH21-:UNK0:O	2.18984
			A:GLN161:HE21-:UNK0:O	2.33823
			A:CYS163:HN-:UNK0:N	2.17761
			A:ARG207:HE-:UNK0:F	2.09034
			:UNK0:H1-A:SER120:O	1.94467
			:UNK0:H1-A:GLN161:O	2.53344
			:UNK0:H2-A:GLY122:O	2.29135
Apoptosis regulator Bcl-2 (PDB:4MAN)	−2.64	11.67mM	A:ARG106:HN-:UNK0:N	2.40095
			A:ARG107:HN-:UNK0:N	2.10937
			A:ASP108:HN-:UNK0:O	1.8044
			:UNK0:H1-A:ASP108:OD2	1.83811
			:UNK0:H2-A:ARG104:O	2.96844
TRAF2 and NCK-interacting protein kinase (PDB: 2X7F)	−3.89	1.40 mM	:UNK0:H1-A:GLU69:OE1	1.67651
			:UNK0:H2-A:LEU169:O	2.01299
			A:ASP171:HN-:UNK0	2.44016
Cyclin-dependent protein kinase-2 (PDB:6GUE)	−5.82	54.62uM	A:LYS33:HZ1-:UNK0:F	2.09629
			A:LEU83:HN-:UNK0:O	2.38077
			A:ASP145:HN-:UNK0:O	2.06687
			:UNK0:O-A:GLU81:O	3.1013
			:UNK0:O-A:PHE80	4.03351

**Table 4 pharmaceuticals-15-01348-t004:** Comparative analysis of data obtained after docking with emodin and chrysophanol and FDA-approved anticancer drug fluorouracil.

Receptor Name	Anticancer Drug Fluorouracil Estimated Free Energy of Binding (Kcal/mol)	Emodin Estimated Free Energy of Binding (Kcal/mol)	Chrysophanol Estimated Free Energy of Binding (Kcal/mol)
Caspase-3 (PDB:4QU8)	−4.10	−7.42	−7.37
Apoptosis regulator Bcl-2 (PDB:4MAN)	−2.64	−6.55	−6.83
TRAF2 and NCK-interacting protein kinase (PDB: 2X7F)	−3.89	−8.15	−8.25
Cyclin-dependent protein kinase-2 (PDB:6GUE)	−5.82	−7.61	−7.71

**Table 5 pharmaceuticals-15-01348-t005:** ADME prediction from SwissADME (GI = Gastrointestinal, BBB = Blood-brain barrier, Pgp = P glycoprotein, CYP = Cytochrome, log Kp = skin permeation).

Compounds	GI Absorption	BBB Permeant	Pgp Substrate	CYP1A2 Inhibitor	CYP2C19 Inhibitor	CYP2C9 Inhibitor	CYP2D6 Inhibitor	CYP3A4 Inhibitor	log Kp (cm/s)
Chrysophanol	High	Yes	No	Yes	No	No	No	Yes	−5.34
Emodin	High	No	No	Yes	No	No	No	Yes	−6.02

**Table 6 pharmaceuticals-15-01348-t006:** Drug-likeness prediction from SwissADME server (MW = Molecular Weight, TPSA = total polar surface area, Consensus Log P = average of all predicted Log Po/w.

Compounds	MW (g/mol)	Rotatable Bonds	H-Bond Acceptors	H-Bond Donors	TPSA (Å^2^)	Consensus Log P	Lipinski Violations	Ghose Violations	Veber Violations	Egan Violations	Muegge Violations	Bioavailability Score	Synthetic Accessibility
Chrysophanol	254.24	0	4	2	74.60	2.38	0	0	0	0	0	0.55	2.47
Emodin	270.24	0	5	3	94.83	1.87	0	0	0	0	0	0.55	2.57

**Table 7 pharmaceuticals-15-01348-t007:** Toxicity prediction. Data obtained from pkCSM server (http://biosig.unimelb.edu.au/pkcsm/theory, accessed on 18 January 2022).

S.No	Compounds	AMES Toxicity	Max. Tolerated Dose (Human)	hERG I Inhibitor	hERG II Inhibitor	Oral Rat Acute Toxicity (LD50)	Oral Rat Chronic Toxicity (LOAEL)	Hepatotoxicity	Skin Sensitization	*T. pyriformis* Toxicity	Minnow Toxicity
1	Chrysophanol	Yes	−0.256	No	No	2.275	2.057	No	No	0.794	1.603
2	Emodin	No	0.158	No	No	2.116	2.074	No	No	0.536	2.057

## Data Availability

Data is contained within the article.

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
