# Peer review of "In Vitro, Molecular Docking and In Silico ADME/Tox Studies of Emodin and Chrysophanol against Human Colorectal and Cervical Carcinoma"

_pharmaceuticals, 2022, doi:10.3390/ph15111348_

Round 1
Reviewer 1 Report
Reviewers comment
In the study titled ‘In vitro, molecular docking and in silico ADME/Tox studies of Emodin and Chrysophanol against human colorectal and cervical carcinoma’ the authors Ahmad et al. have studied possible targets of Anthraquinones (AQs) derivatives by molecular docking. They also studied ADME properties with SwissADME software. The anti-cancer and toxicity studies were done in cell-based assays.
The study is interesting, but there are many studies on pharmacokinetics, toxicity, and a few on molecular docking of Emodin and Chrysophanol. The review of the literature is not satisfactory. The authors could have correlated their computer-based prediction to already reported data in cell lines or in vivo. Here are a few suggestions that might improve the value of the manuscript.
- The authors may consider correlating their computer-based prediction to already reported data in cell lines or in vivo. Here are some articles.
- Emodin: PMID: 27188216, PMID: 32973538, PMID: 33509280, PMID: 34629100, PMID: 34629100, PMID: 31011292
- Chrysophanol PMID: 31373015, PMID: 31655854, PMID: 34572468, PMID: 31710116
2. Line 266: Legends - is this correct?
3. Line 355, 360: BOILED egg graph in Fig 15, but mentioned as Figure 11
4. Capecitabine is a prodrug, converts to active form fluorouracil, by thymidine phosphorylase. Hence is it not ideal to use fluorouracil rather than capecitabine for comparison?
5. The mechanism of action of fluorouracil is to be included. Why is it selected for comparison for molecular interaction (Line 38) or drug-target interactions with Cas3, Bcl2, TNIK, and CDK2?
Author Response
Reviewer’s comment
In the study titled ‘In vitro, molecular docking and in silico ADME/Tox studies of Emodin and Chrysophanol against human colorectal and cervical carcinoma’ the authors Ahmad et al. have studied possible targets of Anthraquinones (AQs) derivatives by molecular docking. They also studied ADME properties with SwissADME software. The anti-cancer and toxicity studies were done in cell-based assays.
The study is interesting, but there are many studies on pharmacokinetics, toxicity, and a few on molecular docking of Emodin and Chrysophanol. The review of the literature is not satisfactory. The authors could have correlated their computer-based prediction to already reported data in cell lines or in vivo. Here are a few suggestions that might improve the value of the manuscript.
Response: Thank you for your constructive suggestions, as suggested by reviewer the present computer-based prediction has been correlated to already reported data in cell lines or in vivo at appropriate places in the introduction and discussion section and are highlighted in yellow color.
Comment 1: The authors may consider correlating their computer-based prediction to already reported data in cell lines or in vivo. Here are some articles.
- Emodin: PMID: 27188216, PMID: 32973538, PMID: 33509280, PMID: 34629100, PMID: 34629100, PMID: 31011292
- Chrysophanol PMID: 31373015, PMID: 31655854 (This paper has been Retracted; therefore, we did not cite it), PMID: 34572468, PMID: 31710116
Response: Thank you for your constructive suggestions, All the references suggested by reviewer has been cited in the manuscript at appropriate places in the introduction and discussion section and are highlighted in yellow color.
Comment 2: Line 266: Legends - is this correct?
Response: Thank you for your concern, the sentence has been modified in then revised manuscript.
Comment 3: Line 355, 360: BOILED egg graph in Fig 15, but mentioned as Figure 11
Response: Correction has been done
Comment 4: Capecitabine is a prodrug, converts to active form fluorouracil, by thymidine phosphorylase. Hence is it not ideal to use fluorouracil rather than capecitabine for comparison?
Response: Thank you for your concern. In the present study, we aimed to investigate the binding affinities of Emodin and Chrysophanol with four major target proteins i.e., Cas3, Bcl2, TNIK, and CDK2. We used Capecitabine because it’s a widely used FDA approved chemotherapeutic drug, to treat breast cancer, gastric cancer and metastatic colorectal cancer. Capecitabine is also used together with docetaxel to treat metastatic breast cancer. Yes, we can also use fluorouracil instead of capecitabine. But our aim was to just compared their binding affinities with target proteins (Caspase-3, Apoptosis regulator Bcl-2, TNIK and CDK2) being responsible for a wide range of malignancies to compare the scoring functions obtained after docking and for innovative possibilities to find new treatment. We did not evaluate the anticancer activity of any known FDA-approved medication in this investigation. Because of this, we can choose any FDA-approved medicine for molecular docking and in silico analysis.
Comment 5: The mechanism of action of fluorouracil is to be included. Why is it selected for comparison for molecular interaction (Line 38) or drug-target interactions with Cas3, Bcl2, TNIK, and CDK2?
Response: Thank you for your concern. Capecitabine/ fluorouracil is an FDA approved chemotherapy medication used to treat breast cancer, gastric cancer and metastatic colorectal cancer. Capecitabine is also used together with docetaxel to treat metastatic breast cancer. The mechanism of Capecitabine/ fluorouracil is well known as its already an FDA approved chemotherapy, so in my opinion there is no need to add the mechanism of action of these drugs. Because in the present study, we have studied the two major bioactive molecules of Rheum emodi i.e., Emodin and Chrysophanol and compared with one of the FDA approved anti-cancerous drug capecitabine to the selected four major cancer-related protein targets (i.e., Caspase-3, Apoptosis regulator Bcl-2, TNIK and CDK2) being responsible for a wide range of malignancies to compare the scoring functions obtained after docking and for innovative possibilities to find new treatment.
Recent research has shown that multitarget treatments outperform single-target therapeutics in terms of efficacy, toxicity, and drug resistance. The ADMET SAR database was used to predict the toxicity profile and pharmacokinetics of the Chr and EM. Furthermore in-silico results were validated by the in-vitro anticancer activity against HCT-116 and HeLa cell lines to determine anticancer effect.
In the drug development, anthraquinones have become an important family of chemicals due to their wide range of biological characteristics. Even though the researchers have conducted extensive research into anthraquinone anticancer properties against various cancer cell lines through in vitro and in vivo assays. Chrysophanol has not been widely studied in colorectal and cervical carcinoma and there is no particular biochemical mechanism has been found to explore binding methods and interaction energies of these anthraquinone analogues through docking studies with these crucial cancer-related protein targets. The selected proteins were important in therapeutic target of colorectal and cervical cancers that’s why we tried to corelate the efficacy of these important anthraquinone analogues against these carcinomas.
The emodin and chrysophanol are the most important anthraquinone analogues. The selected targets eg. NCK-interacting protein kinase (TNIK) is a promising therapeutic target in the treatment of colorectal cancer furthermore, cancer and other disorders associated with aberrant cellular production can be effectively treated by blocking several enzymes that regulate the cell cycle's progression [1]. Caspase 3,8, and 9 expression levels have been proven in previous research to be effective prognostic indicators in tumors of the digestive system, particularly in colorectal cancer [2]. As apoptotic regulators, the BCL-2 family of proteins has been linked to colorectal cancer (CRC) development, progression, and resistance to treatment [3]. It has been suggested that bcl2 may play a vital role in a relatively early stage of cervical tumorigenesis [4]. Emodin also regulated expression and localization of Bcl-2 family proteins during colon cancer apoptosis [5]. We found less literature on anthraquinone analogues binding mechanisms and interaction energy with these important cancer-related protein targets. Thus, we selected these molecular targets to study Emodin and Chrysophanol binding mechanisms and interaction energy with using protein-ligand molecular docking because these targets were shown to be responsible for a broad spectrum of malignancies. As a result, it is extremely desired to investigate the potential of these anthraquinone libraries in order to discover hit/lead-like compounds for future research.
1-Paudel, P.; Shrestha, S.; Park, S.E.; Seong, S.H.; Fauzi, F.M.; Jung, H.A.; Choi, J.S. Emodin Derivatives as Multi-Target-Directed Ligands Inhibiting Monoamine Oxidase and Antagonizing Vasopressin V1Areceptors. ACS Omega 2020, 5, 26720–26731, doi:10.1021/acsomega.0c03649.
2-Asadi M, Shanehbandi D, Asvadi Kermani T, Sanaat Z, Zafari V, Hashemzadeh S. Expression Level of Caspase Genes in Colorectal Cancer. Asian Pac J Cancer Prev. 2018 May 26;19(5):1277-1280. doi: 10.22034/APJCP.2018.19.5.1277. PMID: 29801534; PMCID: PMC6031845.
3-Singh R, Letai A, Sarosiek K (2019) Regulation of apoptosis in health and disease: the balancing act of BCL-2 family proteins. Nat Rev Mol Cell Biol 20(3):175–193. https://doi.org/10.1038/s41580-018-0089-8
4-Shukla S, Dass J, Pujani M. p53 and bcl2 expression in malignant and premalignant lesions of uterine cervix and their correlation with human papilloma virus 16 and 18. South Asian J Cancer. 2014 Jan;3(1):48-53. doi: 10.4103/2278-330X.126524. PMID: 24665447; PMCID: PMC3961868.
5-Saunders, I.T., Mir, H., Kapur, N. et al. Emodin inhibits colon cancer by altering BCL-2 family proteins and cell survival pathways. Cancer Cell Int 19, 98 (2019). https://doi.org/10.11
Reviewer 2 Report
The work has some advantages, but from my point of view it requires corrections before publishing.
Here are some suggestions:
The effects of emodine and chrysophanol on the viability of cancer and non-cancer cells should be compared with the effects of the FDA approved anticancer drug (for example, containing an anthraquinone moiety as the major component) on the same cells
Why did the authors use the concentrations of 5, 10 and 25 µg/ml to study the effect of emodine and chrysophanol on cell viability? There is no dose-effect relationship, especially for the HCT-116 line.
Lines 402-403: The statement: “The inhibitory concentration (IC50) value for EM and Chr was 5 μg/ml for HeLa and HCT-116 cells, respectively” requires clarification and supplementation
The effect of emodine and chrysophanol on non-cancerous HEK-293 cells is shown in Figures 16 and 17, not in Figures 12 and 13
The number of DAPI stained cells in emodine and chrysophanol treated cells is shown in Figure 14B and C, not in Figures 14B and 3C
Some words in Table 1 are illegible
Table 2 is misplaced
Author Response
Comments and Suggestions for Authors
The work has some advantages, but from my point of view it requires corrections before publishing.
Here are some suggestions:
Comment 1: The effects of emodin and Chrysophanol on the viability of cancer and non-cancer cells should be compared with the effects of the FDA approved anticancer drug (for example, containing an anthraquinone moiety as the major component) on the same cells.
Response: Thank you for your concern. In the present study, we aimed to investigate the binding affinities of Emodin and Chrysophanol with four major target proteins i.e., Cas3, Bcl2, TNIK, and CDK2. And in the present study, we didn’t claim that the Emodin and Chrysophanol has better anticancer activity than that of any known FDA approved chemotherapy medication such as Capecitabine/ fluorouracil, which is used to treat breast cancer, gastric cancer and metastatic colorectal cancer. In the present study, we just investigated the effects of emodin and Chrysophanol on the viability of HCT-116 and HeLa cancer cell and compared their cytotoxic effects with that of non-cancerous cell HEK-293. We have taken non-cancerous cell HEK-293 as a control to compare the effects of Emodin and Chrysophanol on both normal as cancerous cell lines. Therefore, we did not feel to compared the efficacy of Emodin and Chrysophanol with that of any known FDA approved chemotherapy medication as they are already in used to treat for variety of cancer.
In the drug development, anthraquinones have become an important family of chemicals due to their wide range of biological characteristics. Even though the researchers have conducted extensive research into anthraquinone anticancer properties against various cancer cell lines through in vitro and in vivo assays. Chrysophanol has not been widely studied in colorectal and cervical carcinoma and there is no particular biochemical mechanism has been found to explore binding methods and interaction energies of these anthraquinone analogues through docking studies with these crucial cancer-related protein targets. The selected proteins were important in therapeutic target of colorectal and cervical cancers that’s why we tried to corelate the efficacy of these important anthraquinone analogues against these carcinomas.
Comment 2: Why did the authors use the concentrations of 5, 10 and 25 µg/ml to study the effect of emodin and Chrysophanol on cell viability? There is no dose-effect relationship, especially for the HCT-116 line.
Response: The dose has seen selected on the basis of some previously reported studies available in the literature. Yes, it’s true but, we did not observe dose dependent activity against HCT-116 line and we have reported whatever we found in this study.
Comment 3: Lines 402-403: The statement: “The inhibitory concentration (IC50) value for EM and Chr was 5 μg/ml for HeLa and HCT-116 cells, respectively” requires clarification and supplementation
Response: Thank you for your concern. It was a typo-error that has been modified in the revised manuscript.
Comment 4: The effect of emodin and Chrysophanol on non-cancerous HEK-293 cells is shown in Figures 16 and 17, not in Figures 12 and 13
Response: Correction has been done.
Comment 5: The number of DAPI stained cells in emodin and Chrysophanol treated cells is shown in Figure 14B and C, not in Figures 14B and 3C.
Response: Correction has been done.
Comment 6: Some words in Table 1 are illegible
Response: Thank you for your king suggestion. Table 1 has been modified.
Comment 7: Table 2 is misplaced
Response: Correction has been done and Table 2 has been incorporated.
Round 2
Reviewer 1 Report
Reviewers comment
pharmaceuticals-1958543-peer-review-v2
Title : In vitro, molecular docking, and in silico ADME/Tox studies of Emodin and Chrysophanol against human colorectal and cervical carcinoma
The authors have addressed/included most of the suggestions for reviewers. However, the authors' response to comment 4 (Capecitabine is a prodrug, converts to active form fluorouracil, by thymidine phosphorylase. Hence is it not ideal to use fluorouracil rather than capecitabine for comparison?) is not convincing.
I appreciate the authors' effort to include FDA-approved anticancer drug for comparison. But is it appropriate to compare a prodrug to an active drug for in-silco affinity or docking? I am of the opinion that an active drug should be compared with an active drug. In the abstract, the authors claim that emodin and chrysophanol show stronger molecular interaction than that of capecitabine. The authors may consider including molecular interaction studies with fluorouracil as it is the active component. Alternatively, there needs reference stating that capecitabine and fluorouracil have comparable molecular interactions in silico.
Author Response
Comments and Suggestions for Authors
Reviewer’s comment: pharmaceuticals-1958543-peer-review-v2
Title: In vitro, molecular docking, and in silico ADME/Tox studies of Emodin and Chrysophanol against human colorectal and cervical carcinoma
The authors have addressed/included most of the suggestions for reviewers. However, the authors' response to comment 4 (Capecitabine is a prodrug, converts to active form fluorouracil, by thymidine phosphorylase. Hence is it not ideal to use fluorouracil rather than capecitabine for comparison?) is not convincing.
I appreciate the authors' effort to include FDA-approved anticancer drug for comparison. But is it appropriate to compare a prodrug to an active drug for in-silico affinity or docking? I am of the opinion that an active drug should be compared with an active drug. In the abstract, the authors claim that emodin and chrysophanol show stronger molecular interaction than that of capecitabine. The authors may consider including molecular interaction studies with fluorouracil as it is the active component. Alternatively, there needs reference stating that capecitabine and fluorouracil have comparable molecular interactions in silico.
Response: I am very thankful to the reviewer for his constructive suggestions and comments. I would like to let you know that we already performed the docking interaction of fluorouracil with target proteins during the first round of comments and we found that there were no differences in docking interaction and binding energy. The molecular interactions and binding energies of the prodrug (Capecitabine) and active drug (Fluorouracil) were almost similar. That’s why we did not incorporate the docking interaction of fluorouracil during the first revision. For your kind perusal herewith, I am sharing the comparative molecular interaction and binding affinity of a prodrug (Capecitabine) and an active drug (Fluorouracil), which is almost the same. If you wish, we can further incorporate the molecular interaction analysis of fluorouracil instead of Capecitabine. I would like to thank you once again for your kind suggestion that will surely improve the quality of the manuscript.
The comparative molecular interaction and binding affinitites of prodrug (Capecitabine) and an active drug (Fluorouracil) can be found in attached file.
